# Analysis of Discordance between Genotypic and Phenotypic Assays for Rifampicin-Resistant *Mycobacterium tuberculosis* Isolated from Healthcare Facilities in Mthatha

**DOI:** 10.3390/pathogens12070909

**Published:** 2023-07-05

**Authors:** Carine Bokop, Lindiwe M. Faye, Teke Apalata

**Affiliations:** 1Division of Medical Microbiology, Department of Laboratory Medicine and Pathology, Faculty of Health Sciences, Walter Sisulu University, Mthatha 5099, Eastern Cape, South Africa; 2Department of Laboratory Medicine and Pathology, Faculty of Health Sciences and National Health Laboratory Services (NHLS), Walter Sisulu University, Mthatha 5099, Eastern Cape, South Africa; fayelindiwe@yahoo.com

**Keywords:** Xpert MTB/RIF, line probe assay, MGIT 960 system, *Mycobacterium tuberculosis*, discrepant results

## Abstract

The study sought to determine the rate of discordant results between genotypic and phenotypic tests for the diagnosis of drug-resistant tuberculosis (DR-TB). Sputum samples and cultured isolates from suspected DR-TB patients were, respectively, analyzed for *Mycobacterium tuberculosis* by Xpert^®^ MTB/RIF (Cepheid, Sunnyvale, CA, USA) and line probe assays (LPA) (Hain, Nehren, Germany). Discrepant rifampicin (RMP)-resistant results were confirmed using BACTEC MGIT960 (BD, New York, NY, USA). Of the 224 RMP-resistant results obtained by Xpert MTB/RIF, 5.4% were susceptible to RMP by LPA. MGIT960 showed a 75% agreement with LPA. The discrepancy was attributed to either heteroresistance or DNA contamination during LPA testing in 58.3% of cases. In 25% of the samples showing agreement in RMP resistance between Xpert MTB/RIF and MGIT960, the discrepancy was attributed to laboratory errors causing false RMP susceptible results with LPA. In 16.7% of the cases, the discrepancy was attributed to false RMP susceptible results with Xpert MTB/RIF. Out of the 224 isolates, susceptibility to isoniazid (INH) by LPA was performed in 73.7% RMP-resistant isolates, of which, 80.6% were resistant. All RMP-resistant isolates by Xpert MTB/RIF were confirmed in 98.5% by LPA if TB isolates were resistant to INH, but were only confirmed in 81.3% if TB isolates were susceptible to INH (*p* < 0.001). In conclusion, laboratory errors should be considered when investigating discordant results.

## 1. Introduction

Tuberculosis (TB) remains an important and one of the largest infectious diseases worldwide. According to the World Health Organization (WHO), there were an estimated 10.6 million TB cases in 2021, among which, 6.7% were reported of people living with human immunodeficiency virus (HIV) [1]. The majority of TB cases have been reported from the region of South East Asia (45%), followed by the African region (23%), and the Western-Pacific region (18%) in the third position [1]. During that same year, the total number of deaths reported by the WHO was 1.6 million, among which, 1.4 million of those deaths were of HIV non-infected individuals and 187,000 deaths were of people living with HIV infection [1]. With the emergence of multidrug-resistant tuberculosis (MDR-TB) and extensively drug-resistant (XDR) strains, as well as the increased rate of direct transmission, TB has now become an even larger threat. As per definition, MDR-TB refers to resistance to isoniazid and rifampicin, with or without resistance to other first-line drugs. However, XDR-TB is defined as resistant to at least isoniazid and rifampicin, as well as to any of the fluoroquinolone and any of the three second-line injectable drugs (amikacin, capreomycin, and kanamycin). The burden caused by MDR-TB is considerable; in 2021, the incidence of MDR rifampicin-resistant TB (RR-TB) was at 450,000 cases worldwide and was associated with 191,000 deaths [1]. In South Africa (SA), TB disease remains among the leading cause of natural deaths caused by a single organism [2]. Up to 360,000 individuals were reported to have developed TB in 2019 in SA, among which the majority, or more than half of them (58%), were living with HIV infection, and 17% of them were reported dead from the disease [3]. In 2015, MDR-TB in South Africa was at 2.08% based on the global estimations [4].

The rapid and accurate laboratory diagnosis of drug-resistant (DR) strains is vital for proper treatment and management, which might also be one of the most active approaches to decreasing the transmission of DR-TB (MDR/XDR-TB). Phenotypic drug susceptibility testing (DST) of *M. tuberculosis* is considered the “gold standard” test; the use of BACTEC MGIT 960 is operational in many laboratories and is considered a rapid liquid phenotypic DST method, but unfortunately requires a long time (4–6 weeks) to report results. The World Health Organization recommends the use of rapid molecular tests Xpert^®^ MTB/RIF assay (Cepheid, Sunnyvale, CA, USA) and, the WHO also endorsed LPA “line probe assays” (Hain Lifescience, Nehren, Germany) for the rapid screening of XDR-TB in MDR-TB patients [5]. Both tests are based on polymerase chain reaction (PCR) amplification, followed by the detection of mutations in its resistance-determining region (RDR). The assay detects mutations in different genes: the *rpoB* gene is detected for RIF resistance, the *kat*G gene for high-level INH resistance, and *inh*A for low-level INH resistance [6]. Nevertheless, in a few cases, Xpert MTB/RIF, LPA, and phenotypic tests may show conflicting INH and RIF-susceptibility results; highly discordant results have been reported for *M. tuberculosis* isolates, carrying specific resistance-conferring mutations for some first-line drugs [7]. Discrepant results among XpertMTB/Rif and LPA usually occur when GeneXpert reports “Rif resistant” and LPA reports “Rif susceptible”. Some factors which may be considered as responsible for discrepancy results include bacterial population (repeated sub-culturing may lead to losing the slow-growing resistant population and false susceptible results), heteroresistance, different growth kinetics, cross-contamination, mixed infections, the growth difficulties of some strains, and the minimum inhibitory concentration (MIC) of some isolates, which is closer to the critical concentration [8]. It was reported that mixed infections might also be an important mechanism underlying the change in drug susceptibility patterns through the presence or absence of antibiotic pressure, which determined the dominant growth of strains of mixed infections [9].

Phenotypic assays (MGIT 960, BD, New York, USA) and genotypic assays (LPA) can provide data on all first-line and second-line drugs; however, Xpert MTB/RIF detects resistance to rifampicin only [10]. Resistance to rifampicin is a key determinant in treatment failure and generally correlates well with MDR-TB as ~85% of rifampicin-resistant clinical *M. tuberculosis* isolates worldwide are also additionally resistant to isoniazid [10].

Drug-resistant TB (DR-TB) emerged in South Africa by the 1980s but was not thought to be a major problem [11]. The Eastern Cape province, which has the third largest population of nearly 7 million people, is the second poorest province in South Africa with the second lowest rate (10.2%) of individuals with access to medical coverage in the country [12]. In 2010, The Eastern Cape was the second worst-affected province with drug-resistant TB after KwaZulu-Natal with more than 10% of strains having resistance to at least one drug [12]. Therefore, the present study was conducted to investigate discrepancies in clinical *M. tuberculosis* isolates, using phenotypic and molecular methods in patients attending Gateway Clinic in Mthatha, South Africa.

## 2. Materials and Methods

All patients were enrolled following an informed consent.

### 2.1. Study Design

This study was an observational descriptive study. Samples were prospectively collected for 27 months.

### 2.2. Study Setting

Located in a rural area of Mthatha, Gateway Clinic in KSD has a satellite TB clinic that receives and manages all cases suspected of being drug-resistant tuberculosis (DR-TB) from KSD and Mhlontlo, which are the 2 sub-districts where the study took place. Medical records from all DR-TB cases are also kept at Gateway Clinic while all samples are sent to Mthatha at the National health laboratory service (NHLS) TB laboratory for analysis. On its own, Gateway Clinic manages 57 primary health care (PHC) facilities: 33 facilities from KSD and 24 PHC facilities from Mhlontlo. Gateway Clinic performs the following functions: initiation of treatment, monthly monitoring and follow-up, radiology services at Mthatha General, ECG at Mthatha General, mobile injection team and DOTS—Mthatha Hospice, and conducting DR-TB reviews on Wednesdays and Thursdays.

### 2.3. Sample Collection, Transport, and Storage

Sputum samples were prospectively collected from the study participants between June 2015 and September 2017. Sputum samples were collected by patients themselves after receiving instructions from healthcare workers whilst other samples such as CSF and pleural fluids were collected by the attending medical doctors as part of the patients’ routine management, and all samples were transported to the NHLS microbiology laboratory. The large majority of specimens received for the diagnosis were sputum samples. Study specimens were collected in a separate, well-ventilated room or preferably outdoors.

Samples were transported to the laboratory daily, as soon as they were collected. If specimens cannot be transported immediately after collection, they were kept refrigerated (but not frozen) for 2–3 days without significantly affecting the positivity rate of smear microscopy. To avoid the growth of contaminants if the delay exceeds 3 days, an equal volume of cetyl pyridinium chloride (CPC; solution of 1% CPC in 2% sodium chloride) was therefore added.

Unprocessed sputum specimen was stored at 4 °C for a duration of up to 10 days, whilst after the addition of SR buffer to the specimen, they were stored at 2–8 °C for a limited time of 8 h.

To protect the privacy and confidentiality of the patients, no names were recorded; instead, a personalized research number was used for each patient and only the main investigator had access to the collected data.

### 2.4. Laboratory Diagnosis of MDR-TB and XDR-TB

#### 2.4.1. Molecular Diagnosis

##### GeneXpert MTB/RIF

The GeneXpert was performed directly on TB samples using the G4 version of cartridges according to the manufacturer’s recommendation (Cepheid, Sunnyvale, CA, USA). Samples were decontaminated and reagent buffer containing NaOH and isopropanol was added at a ratio of 2:1, followed by incubation at room temperature for 15 min. Two milliliters of the final samples were then transferred into the Xpert MTB/RIF cartridge, and after mixing, the cartridge was loaded into the GeneXpert instrument. The software automatically filled in the reagent lot ID, cartridge number, and expiration date. The results were usually generated after 90–120 min and were recorded. Results were reported as *M. tuberculosis* negative or positive, and RMP resistant or susceptible.

##### Line Probe Assay: GenoType MTBDRplus

Molecular diagnosis was also conducted using line probe assay (LPA) and was performed according to the manufacturer’s instructions (Hain Lifescience, Nehren, Germany). Line probe assays are tests that use PCR and reverse hybridization methods for the rapid detection of mutations associated with drug resistance. Line probe assays are designed to identify *M. tuberculosis* complex and simultaneously detect mutations associated with drug resistance for first- and second-line anti-TB agents. One type of LPA is the GenoType MTBDR*plus* and was designed for simultaneous detections of the most important *rpoB* mutations which confer RMP resistance, as well as *kat*G and *inh*A mutations, which confer high- and low-level INH resistance, respectively. Other genes included: *rrs* for kanamycin/amikacin (Km/Am) resistance; *tly*A for capreomycin (Cm) resistance; and *gyr*A/*gyr*B for moxifloxacin/ofloxacin (Mfx/Ofx) resistance. Mutations associated with MDR in DNA are mostly located on a specific region known as rifampicin resistance-determining region (RRDR) of *rpoB*. Genetic alteration within an 81 bp rifampicin resistance-determining region (RRDR) can cause up to 95% of rifampicin resistance [13]. Primers, probes, and their oligonucleotide sequences were selected from Table 1.

The LPA test is based on DNA strip technology and has four consecutive steps, including DNA extraction, amplification, hybridization, and, the evaluation/interpretation of results as described in the picture below. All four steps were performed as per the WHO recommendations. Briefly, DNA was extracted from sputum or cultivated sample using the GenoLyse kit. Next, multiplex PCR amplification of the resistance-determining region of the gene under question was then performed using biotinylated primers, and the master mix. Following amplification, labeled PCR products were hybridized with specific oligonucleotide probes immobilized on a strip. If a mutation was present in one of the target regions, the amplicon will not hybridize with the relevant probe. After extraction and PCR amplification of the resistance-determining region of DNA, mutations were detected by the presence or absence of binding to “probes”, indicated by the presence or absence of colored bands on a strip. The assay was performed and the results were interpreted according to the manufacturer’s recommendations. Details of each step are described below.

a.DNA Extraction

Decontaminated smear-positive or -negative sputum samples as well as mycobacteria grown in solid medium or liquid medium were used as starting material for DNA extraction. The GenoLyse kit was used for DNA extraction from decontaminated clinical specimens or cultures as seen in Figure 1 (DNA extraction). Each DNA extracted from sputum or a cultivated sample using the GenoLyse kit was used for amplification with the GenoType MTBDRsl VER 2.0 kit.

b.Multiplex PCR Amplification

All reagents needed for amplification such as polymerase and primers were already provided by the manufacturer and included in the amplification mixed A and B (AM-A and AM-B) (Figure 1). After thawing, AM-A and AM-B were spanned down briefly and mixed carefully by pipetting up and down; each sample was prepared as follows: 10 µL AM-A, 35 µL AM-B, and 5 µL of DNA solution. The number of samples to be analyzed plus control samples were determined and prepared. A master mix containing AM-A and AM-B was prepared and mixed carefully but thoroughly (without using a vortex). The content of the AM-A reaction tube was transferred into the AM-B reaction tube; this led to a 0.68 mL master mix for 12 amplification reactions. An aliquot of 45 µL into each of the prepared PCR tubes was prepared and 5 or 10 µL water was added to one aliquot (negative control), and 5 or 10 µL DNA solution was added to each aliquot (except for negative control). A thermal cycler was used and “MDR DIR” was selected for clinical specimens; otherwise, the protocol “MDR CUL” was selected for cultivated samples.

### 2.5. Phenotypic DST

The automated BACTEC mycobacterial growth indicator tube (MGIT) 960 (Becton Dickinson, New York, NY, USA) is an in vitro diagnostic instrument designed and optimized for the rapid detection of mycobacteria from clinical specimens (except blood) or from grown isolates. All sections of the MGIT are shown in Figure 2A. It provides continuous monitoring of patient samples to identify the positive ones and the non-positive ones. Diagnosing TB using culture takes on average of four weeks to obtain a definitive test result using solid media, with another four to six weeks to then obtain the drug susceptibility results. Meanwhile, the patient is in need to start their TB drug treatment. Therefore, one way to reduce this delay has been to use liquid media, such as BACTEC 960. This system has a 960-tube capacity for nearly 8000 specimens per year and is useful in laboratories dealing with large specimen loads. The MGIT 960 tube contains 7.0 mL of modified 7H9 liquid broth base; the MGIT 960 growth supplement is added to the tubes together with the MGIT PANTA, a mixture of antibiotics used to reduce contamination. A volume of 0.5 mL of a well-mixed processed/concentrated specimen is then added to the appropriately labeled MGIT tube and mixed thoroughly (Figure 2B). During bacterial growth within the tube, oxygen is used and is replaced with carbon dioxide. With the depletion of free oxygen, the fluorochrome is no longer inhibited, resulting in fluorescence within the MGIT tube; the intensity of fluorescence is directly proportional to the extent of oxygen depletion. The results indicating susceptibility or resistance are interpreted and reported automatically by the MGIT system using predefined algorithms that compare bacterial growth in the drug-containing tube with the growth in the drug-free control tube (Becton, Dickinson and Company). The growth unit (Gu) values of the drug-containing vials were evaluated. In order to interpret the results, when growth control (GC) reached the value of 400 or more within 3 to 13 days, the instrument indicated that the test was complete. After scanning different tubes, an inventory report was printed, and results for INH, RIF, Amk, Kan, Cm, Ofx, and Lfx were interpreted by the instrument as “S” for susceptible or “R” as resistant. When the GU of the drug-containing tube was <100, the result was reported as susceptible and when the GU of the drug-containing tube was ≥100, the result was reported as resistant strains. To ascertain that results were truly susceptible when GU was initially found to be <100, the test tube was incubated for a further 7 days, and if it was still <100, the strain was then reported as “true susceptible”.

### 2.6. Data Analysis

Statistical analysis was performed using SPSS version 22.0. Continuous variables were expressed as mean ± standard error of the mean (SEM) and categorical variables were expressed as proportions (%). Student *t*-test was used to compare discrepant results between phenotypic and genotypic methods. The results were reported as statistically significant when the *p*-value was ≤0.05.

## 3. Results

Between June 2015 and September 2017, 224 patients suspected of having DR-TB were transferred from clinics located in KSD and Mhlontlo sub-districts to Gateway MDR-TB clinic in Mthatha, Eastern Cape. GeneXpert^®^ MTB/RIF assay (Cepheid, USA) was performed on sputum samples obtained from those patients, and the presence of *Mycobacterium tuberculosis* with mutations in *rpoB* genes (resistance to RMP) was determined in all suspected 224 cases.

Following cultures, LPA tests were performed on 165 initial TB isolates (out of 224), of which, 133/165 (80.6%) isolates were resistant to INH and 157/165 (95.2%) were resistant to RMP. When classified by INH susceptibility results, all 165 RMP-resistant isolates by Xpert MTB/RIF were confirmed in 98.5% by LPA if *M. tuberculosis* isolates were resistant to INH, but only confirmed in 81.3% if *M. tuberculosis* isolates were susceptible to INH (*p* < 0.001) as described in Table 2.

At the end of the study, out of the 224 RMP-resistant TB samples by Xpert MTB/RIF, 12 (5.4%) were found to be susceptible to RMP by LPA. Table 3 below shows the demographic, clinical, and outcomes information of the 12 participants whose samples displayed discrepant RMP results between LPA and Xpert^®^ MTB/RIF.

The 12 discrepant samples were further subjected to MGIT 960 drug susceptibility testing. The MGIT 960 results showed a 75% agreement with LPA results. Table 4 describes the 12 discrepant isolates of *M. tuberculosis* and provides possible reasons for the occurrences of discrepancies between Xpert MTB/RIF, LPA, and MGIT 960. The discrepancy was attributed to either heteroresistance (mixed infections) or DNA contamination during LPA testing in seven (58.3%) of the twelve discrepant cases. In three (25%) other samples that showed agreement in RMP resistance between Xpert MTB/RIF and MGIT960 (but disagreement with LPA), the discrepancy was attributed mostly to laboratory error causing false RMP susceptible results with LPA (either sample mixed up or DNA contamination). For the remaining two (16.7%) cases, the discrepancy was attributed mostly to laboratory errors causing false RMP susceptible results with Xpert MTB/RIF (either Xpert readout errors with a Ct value of 4.1–4.9 or a sample transport delay).

## 4. Discussion

According to the WHO, an effective treatment regimen depends on an optimal susceptibility testing of *Mycobacterium tuberculosis* to anti-TB drugs [4]. Over the past years, new technologies have been introduced to shorten and improve methods for the detection of anti-TB drug resistance, notably at molecular and phenotypic levels. As the use of such molecular and phenotypic assays increases, discordance between results has been encountered.

We had cases with discrepancy results between LPA and GeneXpert when testing for rifampicin resistance in our study. In many cases, discrepancies often occur due to: bacterial population (repeated sub-culturing may lead to losing the slow-growing resistant population and false susceptible result), heteroresistance, cross-contamination, mixed infections, the growth difficulties of some strains, and the minimum inhibitory concentrations (MIC) of some isolates, which are closer to the critical concentration [8].

In this present study, we found that the rifampicin discrepancy rate between GeneXpert and LPA was very low (1.5%) if INH was subsequently resistant, whilst a very high discrepancy rate (19.7%) was observed if INH was subsequently susceptible (*p* < 0.001). Thus, we hypothesized that susceptibility to INH might also play a role in the rifampicin discrepancy results between GeneXpert and LPA. However, further studies with a large number of participants are required to draw a conclusion.

In this present study, from the 12 discrepant results identified, three main reasons for discordance were observed:1.Technical laboratory errors causing false resistant RMP results with GeneXpert was considered because RMP resistance was identified only on Xpert MTB/RIF while LPA and MGIT 960 indicated RMP susceptibility.

In this study, we used Xpert MTB/RIF version G4 of cartridges (Cepheid, USA) that was introduced to reduce false RMP-resistant results as compared to the G3 version. The Xpert MTB/RIF assay detects *M. tuberculosis* and RMP resistance by PCR amplification of the rifampin resistance-determining region (RRDR) of the *M. tuberculosis rpoB* gene and the subsequent probing of this region for mutations that are associated with RMP resistance. Although the study by Helb et al. established that the Xpert MTB/RIF assay has a limit of detection (LOD), defined as the minimum number of bacilli that can be detected with 95% confidence of 131 CFU per ml of clinical sputum, this was not observed in our study [14]. Of our twelve discrepant samples (all with positive Xpert MTB/RIF results), five had negative smear microscopy results and one sample had scanty AFB observed. The remaining six samples had moderate to many AFB observed. General troubleshooting for the presumed laboratory errors was performed and included a verification of samples, a review of the used technique that could allow the occurrence of cross-contamination, reagents’ quality control, Xpert read-out errors, and either a dropout or delay in probes’ hybridization by ensuring that we did not have ΔCt value between 4.1 and 4.9 on Xpert. Drop-out was excluded since none of the probes had a Ct value of zero, and the delta Ct max value for each test was not between 4.1 and 4.9, hence Xpert readout errors were excluded. Although cross-contamination was not fully excluded, sample mix-up was likely the reason for the discrepancy in our two results out of twelve. Cross-contamination is very rare with Xpert MTB/RIF since it operates on a closed system.

2.Heteroresistance was considered since there was a simultaneous presence of all rpoB wild types (wts) and specific rpoB mutation signals in LPA in the presence of RMP susceptible by MGIT 960 but RMP resistant on Xpert MTB/RIF

We identified possible 7 (3.1%) out of 224 cases of heteroresistance in this study; those are usually defined as the coexistence of susceptible and resistant *M. tuberculosis* strains in the same patient [15]. A similar result was found in a study conducted in Ethiopia by Mekonnen and his colleagues, reporting eight (1.9%) cases of RMP heteroresistance [16]. In our study, the level of heteroresistance was lower compared to studies conducted in Uzbekistan (South Central Asian Union) by Hofmann-Thiel and colleagues with 20% of heteroresistance, and another study conducted in India with 34% of heteroresistance cases [15,17].

Heteroresistance might be the result of several factors such as the presence of both resistant and susceptible Mtb isolates, or the endogenous development of two sub-populations of Mtb isolates after inadequate treatment, since in the same patient with different drug susceptibility patterns, several sub-populations may co-exist [18]. It is also believed that heteroresistance develops during the treatment of DR-TB [19]. Since we had many patients on anti-TB treatment and also had a history of previous exposure to anti-TB drugs, the possibility of endogenous development of sub-populations was therefore raised among our cohort of patients who displayed heteroresistance results.

The direct transmission of heteroresistance of both susceptible and resistant bacterial populations from drug-resistant patients to previously untreated cases could also have occurred. Heteroresistance has been proven so far to occur mostly in high TB-incidence locations. For a country such as South Africa where the prevalence of DR-TB strains is very high, the level of patients directly infected with resistance is therefore high. In different parts of the world, heteroresistance was found to be low at 1.4% in Italy [20], 1.9% in Russia [21], and 1.9% in Pakistan [22]. The possibility of DNA sample contamination during LPA tests was also investigated but could not be fully excluded.

3.Technical laboratory errors causing false susceptible RMP results with LPA since RMP resistance in Xpert was confirmed using MGIT 960 performed on positive cultures (while LPA indicated discordant results).

Molecular techniques such as LPA have revolutionized the diagnosis of DR-TB. The contamination of DNA samples during LPA testing is not an uncommon phenomenon observed in LPA laboratories. Factors such as laboratory air and surfaces, tools, and equipment are potential sources for contaminating DNA during a pre-LPA testing procedure [23]. More sources of DNA contamination could also be molecular biology grade water, LPA reagents, and DNA extraction kits since they have all been reported as major sources of DNA contamination by many studies [24,25,26]. Nevertheless, contamination during LPA can also be caused by the direct transfer of contaminating DNA from an analyst or any person in the laboratory to the sample ready for LPA testing. DNA contamination can also be due to an object used in the premise of the laboratory and, afterward, from this object to the sample. Therefore, laboratory and personal equipment may consequently act as a vector for DNA contamination.

In a TB laboratory, it is essential for personal protective equipment (PPE) to be applied appropriately, hence equipment such as masks, hats, gloves, and lab coats are worn to prevent contamination. However, unfortunately, if not appropriately used, PPE could also be a vector for DNA contamination.

In addition to possible DNA sample contamination, troubleshooting also included the verification of sample identities to exclude sample mix-up cases. Since our laboratory receives many samples for possible diagnoses of DR-TB, there is a high workload with the possibility of samples being mixed up, resulting in discrepant results being reported.

## 5. Conclusions

Discordance between genotypic and phenotypic tests is increasingly recognized and is becoming a concern particularly in a country such as South Africa, where the incidence of tuberculosis is still high. For first-line anti-TB agents, the discrepancy between LPA and GeneXpert was significantly associated with INH susceptibility. Laboratory errors such as sample mix-up and LPA contamination as well as cases of heteroresistance were among the predominant reasons for discrepant results between the two genotypic tests and the used phenotypic method. Findings from this present study are important for regional TB control program managers, who need to doubly evaluate the performance of the Xpert MTB/RIF before utilizing them in DR-TB control programs.

Limitations for this study include: (i) a small number of discordant results; (ii) the absence of a cohort of susceptible TB for comparison with DR-TB since we worked in a DR-TB clinic only; and (iii) the loss of some clinical and/or laboratory findings that led to the exclusion of some cases during analysis.

## Figures and Tables

**Figure 1 pathogens-12-00909-f001:**
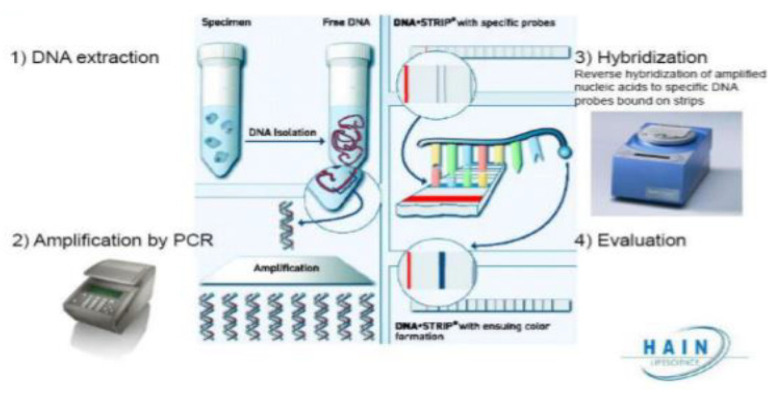
HAIN genotype line probe assay (LPA) (https://media.tghn.org/medialibrary/2020/09/Hain_GenoType_Line_Probe_Assay_Pacome_Abdul.pdf, accessed on 21 June 2023).

**Figure 2 pathogens-12-00909-f002:**
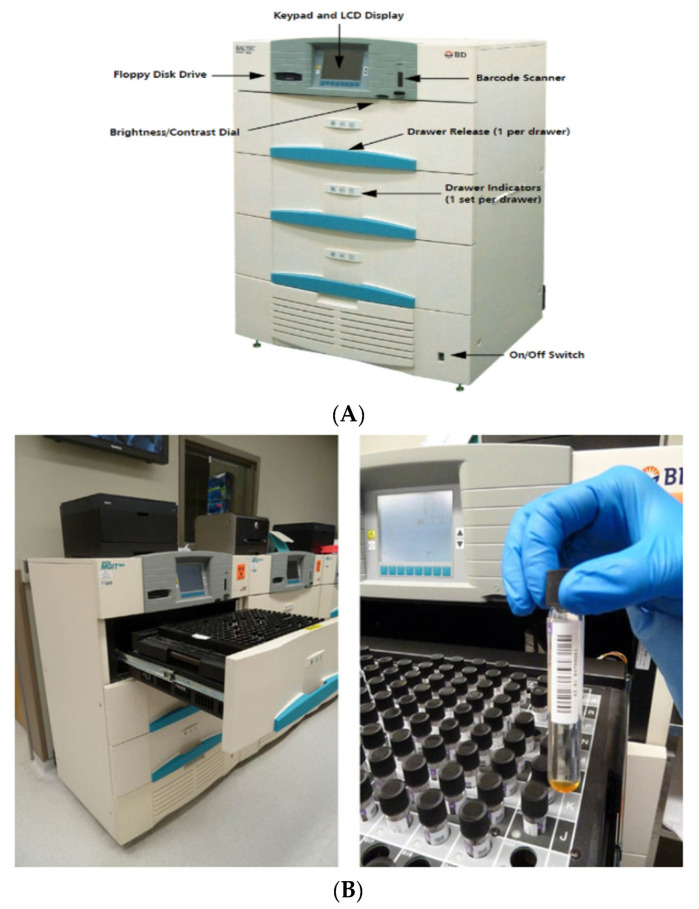
(**A**) The BACTEC™ MGIT™ 960 instrument (Becton Dickinson) and culture tubes for broth-based mycobacterial growth: https://ntep.in/node/1464/CP-mgit-960-instrument-parts (accessed on 21 June 2023); (**B**) The BACTEC™ MGIT™ 960 instrument (Becton Dickinson) and culture tubes for broth-based mycobacterial growth: https://www.researchgate.net/figure/The-BACTEC-MGIT-960-instrument-Becton-Dickinson-and-culture-tubes-for-broth-based_fig2_303560156 (accessed on 21 June 2023).

**Table 1 pathogens-12-00909-t001:** Nucleotide sequences used for DNA sequencing of RRDR and complete *rpoB* genes [13].

Primer	Type	Sequence (5′→3′)	Position in Reference to Start Codon (bp)
TR8x	PCR sequencing primer (RRDR)	TCGCCGCGATCAAGGAGTTCTTCGGC	1253 to 277
TR9x	PCR sequencing primer (RRDR)	TGCACGTCGCGGACCTCCAGCCCGGCAC	1382 to 1409
*rpoB*-pcrF	PCR sequencing primer (complete *rpoB*)	CTTTGTTCGTGGTGAGCGTGAG	−565 to −543
*rpoB*-pcrR	PCR sequencing primer (complete *rpoB*)	GTTGATCGTCTCCGGCTTTTTG	3647 to 3668
*rpoB*-2F	sequencing primer (complete *rpoB*)	TGCGGCTCAGCGGTTTAG	−147 to −130
*rpoB*-3R	sequencing primer (complete *rpoB*)	GCCGCGATAATTTTGTCGG	−88 to −70
*rpoB*-4F	sequencing primer (complete *rpoB*)	GTCGACGAGTGCAAAGACAAGG	337 to 358
*rpoB*-5R	sequencing primer (complete *rpoB*)	GAAGTCACCCATGAACACCGTC	432 to 453
*rpoB*-6F	sequencing primer (complete *rpoB*)	CGAGCCCCCGACCAAA	834 to 849
*rpoB*-7R	sequencing primer (complete *rpoB*)	TGCAGCCCGAGCTTCTTG	933 to 950
*rpoB*-8F	sequencing primer (complete *rpoB*)	CCCGATCGAAACCCCTGA	1434 to 1451
*rpoB*-9R	sequencing primer (complete *rpoB*)	GCGAGCCGATCAGACCGA	1463 to 1480
*rpoB*-10F	sequencing primer (complete *rpoB*)	GTGATGCACGACAACGGCA	1960 to 1978
*rpoB*-11R	sequencing primer (complete *rpoB*)	CGCATCCGGTAGGTACGC	1983 to 2000
*rpoB*-12F	sequencing primer (complete *rpoB*)	GGTGAGACCGAGCTGACGC	2410 to 2428
*rpoB*-13R	sequencing primer (complete *rpoB*)	GCGTCAGCTCGGTCTCACC	2410 to 2428
*rpoB*-14F	sequencing primer (complete *rpoB*)	ACGGCAAGGCCATGCTCTTC	2996 to 3015
*rpoB*-15R	sequencing primer (complete *rpoB*)	TGTAGGCAGCACCGTAGGC	3214 to 3232

**Table 2 pathogens-12-00909-t002:** Agreements between Xpert MTB/RIF and LPA by INH susceptibility results (*n* = 165).

Drug Susceptibility Profile	RMP Resistant by GeneXpert *n* (%)	RMP Resistant by LPA *n* (%)	*p*-Value
LPA			<0.001
INH susceptible (*n* = 32)	32 (100)	26 (81.3)	
INH resistant (*n* =133)	133 (100)	131 (98.5)	

RMP: Rifampicin; INH: isoniazid.

**Table 3 pathogens-12-00909-t003:** Demographic, clinical, and outcomes information of the 12 participants whose samples showed discrepancies between LPA and Xpert^®^ MTB/RIF.

Patient No.	Gender	Age	Sub-District of Origin	HIV Status	TB History	Final Diagnosis	Outcomes
1	F	34	KSD	Neg	New	MDR	Still on anti-TB
2	M	66	KSD	Pos	PT	MDR	Still on anti-TB
3	F	28	KSD	Neg	PT	INH-mono	Fav outcome
4	M	35	KSD	Neg	New	MDR	Fav outcome
5	F	55	KSD	Pos	PT	MDR	Still on anti-TB
6	F	30	KSD	Pos	PT	MDR	Poor outcome
7	F	40	KSD	Neg	PT	MDR	Fav outcome
8	F	42	KSD	Pos	New	MDR	Fav outcome
9	F	22	KSD	Pos	New	MDR	Fav outcome
10	M	18	Mhlontlo	Neg	New	Pre-XDR	Fav outcome
11	F	39	KSD	Neg	PT	MDR	Fav outcome
12	M	68	KSD	Neg	New	RMP-mono	Poor outcome

F = Female, M = male, Neg = HIV negative, Pos = HIV positive, PT = previously treated, ST = still on treatment, Fav = favourable, PO = poor outcomes, MDR = multiple drug-resistant.

**Table 4 pathogens-12-00909-t004:** Analysis of discordance between genotypic and phenotypic assays among the 12 DR-TB cases.

Case No.	AFB Smear Results	Xpert MTB/RIF (On Sputum)	LPA (On Isolates)	MGIT-960 (On Isolates)	Possible Reasons for Discordant Results	Possible Errors That Were Investigated
1.	Negative	Resistant	Susceptible (All wild types are present, no *rpoB* mutation signal is present)	Susceptible	Most likely laboratory error causing false resistant RMP result with GeneXpert OR Erroneous Xpert RMP result	Sample mix up DNA contamination for Xpert Xpert readout errors Delay in probes’ hybridization Ct value 4.1–4.9
2.	3+	Resistant	Susceptible (All wild types are present, no *rpoB* mutation signal is present)	Susceptible	Most likely laboratory error causing false resistant RMP result with Gene Xpert OR Erroneous Xpert RMP result	Sample mix up DNA contamination for Xpert Xpert readout errors Delay in probes’ hybridization Ct value 4.1–4.9
3.	2+	Resistant	Susceptible (All wild types are present, *rpoB* mutation present)	Susceptible	Heteroresistance (presence of both resistant and susceptible Mtb isolates; or endogenous development of two sub-populations of Mtb isolates after inadequate treatment) OR DNA contamination for LPA	DNA contamination of LPA
4.	Negative	Resistant	Susceptible (All wild types are present, *rpoB* mutation present)	Susceptible	Heteroresistance (presence of both resistant and susceptible Mtb isolates; or endogenous development of two sub-populations of Mtb isolates after inadequate treatment) OR DNA contamination for LPA	DNA contamination of LPA
5.	Negative	Resistant	Susceptible (All wild types are present, *rpoB* mutation present)	Susceptible	Heteroresistance (presence of both resistant and susceptible Mtb isolates; or endogenous development of two sub-populations of Mtb isolates after inadequate treatment) OR DNA contamination for LPA	DNA contamination of LPA
6.	Negative	Resistant	Susceptible (All wild types are present, *rpoB* mutation present)	Susceptible	Heteroresistance (presence of both resistant and susceptible Mtb isolates; or endogenous development of two sub-populations of Mtb isolates after inadequate treatment) OR DNA contamination for LPA	DNA contamination of LPA
7.	3+	Resistant	Susceptible (All wild types are present, *rpoB* mutation present)	Susceptible	Heteroresistance (presence of both resistant and susceptible Mtb isolates; or endogenous development of two sub-populations of Mtb isolates after inadequate treatment) OR DNA contamination for LPA	DNA contamination of LPA
8.	Negative	Resistant	Susceptible (All wild types are present, *rpoB* mutation present)	Susceptible	Heteroresistance (presence of both resistant and susceptible Mtb isolates; or endogenous development of two sub-populations of Mtb isolates after inadequate treatment) OR DNA contamination for LPA	DNA contamination of LPA
9.	3+	Resistant	Susceptible (All wild types are present, *rpoB* mutation present)	Susceptible	Heteroresistance (presence of both resistant and susceptible Mtb isolates; or endogenous development of two sub-populations of Mtb isolates after inadequate treatment) OR DNA contamination for LPA	DNA contamination of LPA
10.	1+	Resistant	Susceptible (All wild types are present, no *rpoB* mutation is present)	Resistant	Most likely laboratory error caused false susceptible RMP results with LPA	Sample mix up DNA contamination of LPA
11.	3+	Resistant	Susceptible (All wild types are present, no *rpoB* mutation is present)	Resistant	Most likely laboratory error caused false susceptible RMP results with LPA	Sample mix up DNA contamination of LPA
12.	2+	Resistant	Susceptible (All wild types are present, no *rpoB* mutation present)	Resistant	Most likely laboratory error caused false susceptible RMP results with LPA	Sample mix up DNA contamination of LPA

AFB: Acid-fast bacillus, LPA: line probe assay, Mtb: *Mycobacterium tuberculosis,* DNA: deoxyribonucleic acid, Ct: cycle threshol.

## Data Availability

Data can be requested from the corresponding author.

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
