# Peer review of "Analysis of Discordance between Genotypic and Phenotypic Assays for Rifampicin-Resistant Mycobacterium tuberculosis Isolated from Healthcare Facilities in Mthatha"

_pathogens, 2023, doi:10.3390/pathogens12070909_

Round 1
Reviewer 1 Report
The study aims to determine the rate of discordant results between genotypic and phenotypic tests for DR-TB diagnosis. The results present an interesting observation wherein all RMP resistant isolates by Xpert were confirmed in 98.5% of samples by LPA if the isolates were INH resistant. However, the sample size in the case of INH-susceptible samples was low.
The authors should indicate the procedure undertaken in a situation where discrepant results are observed: Are samples recollected from the patients for repeat testing? Is the phenotypic result considered as the true result?
The key takeaway from this manuscript appears to be the highlighting of possible laboratory errors leading to discrepant results. The authors should refer to databases to identify if any previous reports of discrepant results between genotypic and phenotypic results due to laboratory results have been reported. The discussion/conclusion should highlight if laboratory errors appear to be a reason for invalid results in other regions in South Africa and should also highlight the risks associated with it (delayed diagnosis and treatment, risk of further spread, risk of mortality to the patient, etc).
Although overall well-structured, English language needs to be extensively corrected throughout the manuscript.
Line 34: Remove of and replace among with of
Line 36: Should be African region instead of Africa region
Line 38: should be number of deaths instead of death; Replace from with by
Line 48: Replace to with with; Country name should be Capital (South Africa)
Line 57: replace remain with is
Line 85: Remove has
Line 86: Replace has with with
Line 106: Replace was with is
Line 112: Replace his with its
Line 130: Replace method with diagnosis; replace done with conducted
Check entire manuscript for errors in capital vs non-capital
Author Response
See responses uploaded

Reviewer 2 Report
attached

attached
Author Response
See responses uploaded

Round 2
Reviewer 2 Report
none.